

# A Simple Method for Retrieving Significant Wave Height from Dopplerized X-Band Radar

Ruben Carrasco, Michael Streßer, and Jochen Horstmann

Department of Radar Hydrography, Institute of Coastal Research, Helmholtz-Zentrum Geesthacht, Germany

*Correspondence to*: Michael Streßer (michael.stresser@hzg.de)

**Abstract.** Retrieving spectral wave parameters such as the peak wave direction and wave period from marine radar backscatter intensity is very well developed. However, the retrieval of significant wave height is difficult because the radar image spectrum (a backscatter intensity variance spectrum) has to be transferred to a wave spectrum (a surface elevation

variance spectrum) using a modulation transfer function (MTF) which requires extensive calibration for each individual radar setup. In contrast to the backscatter intensity, the Doppler velocity measured by a coherent radar is induced by the radial velocity of the surface scattering and its periodic component is mainly the contribution of surface waves. Therefore, the variance of the Doppler velocity can be utilized to retrieve the significant wave height. Analysing approximately 100 days of Doppler velocity measurements of a coherent on receive radar operating at X-band with vertical polarization in

transmit and receive, a simple relation was derived and validated to retrieve significant wave heights. Comparison to wave measurements of a wave rider buoy as well as an acoustic wave and current profiler resulted in a root mean square error of 0.24 m with a bias of 0.08 m. Furthermore, the different sources of error are discussed and investigated.

## 1 Introduction

Ocean surface waves are one of the most important maritime parameters that are frequently monitored for purposes of

coastal protection, shipping as well as off shore industry operations. Today, surface waves are typically measured by wave gauges from fixed platforms or moored buoys. In order to measure waves from moving platforms or from greater distance, e.g. coastal stations or offshore platforms marine X-band radars have been shown to provide images of ocean surface waves and have therefore been used for measurements of several characteristic wave properties (Young et al. 1985).





The radar backscatter at moderate incidence angles (20° to 80°) is primarily caused by Bragg scattering, a scattering mechanism where the electromagnetic waves couple to small scale surface roughness (~3 cm for X-band) that are aligned with the look direction of the radar. Towards low grazing incidence angles (>85°) additional scattering mechanisms e.g. wedge scattering (Lyzenga et al., 1983) and scattering from micro breakers (Wetzel, 1990), become more and more relevant.

Ocean surface waves are imaged by marine radars because they modulate the small scale surface roughness. The major modulation mechanisms are tilt modulation due to changing surface slopes and hydrodynamic modulation due to the orbital motion of the waves (Alpers et al., 1981). At grazing incidence shadowing modulation becomes of major importance, and it is caused by the very low radar backscatter coming from diffraction in the geometrically shadowed areas of the waves (Barrick, 1995; Plant and Farqueson 2012).

In recent years, X-band marine radars have been utilized to measure spectral wave parameters (Nieto-Borge et al., 1999), wave groups (Dankert et al., 2003), individual waves (Dankert et al., 2004; Nieto-Borge et al., 2004), surface currents (Senet et al., 2001; Huang et al., 2016), bathymetry (Senet et al., 2008; Bell and Osler, 2011) as well as surface winds (Dankert and Horstmann, 2007; Vicen-Bueno et al., 2013). However, to retrieve significant wave heights, the relative radar image spectrum has to be transferred to a real wave amplitude spectrum using a modulation transfer function (Nieto-Borge et al. 1999). A major disadvantage of this method is the inherent need for an extensive calibration of each single radar installation using an additional wave-measuring sensor (Vincent Bueno et al., 2012)

Coherent marine radar systems allow to analyze the Doppler frequency shift of the electromagnetic waves and therefore offer the possibility to calculate the speed of the scattering elements in addition to the backscattered power. Only a few studies exist on the ability to retrieve wave field information from coherent radar measurements of the sea surface. Hwang et al.

(2010) discussed a method to retrieve significant wave heights from space-time Doppler records of the ocean surface with an upwind pointing radar antenna. They suggested an empirical relationship, $H_s = 4\,X\,u_{DRMS}/\omega_p$, where $u_{DRMS}$ is the root mean square value of the Doppler velocity, $X$ an empirical coefficient and $\omega_p$ the peak wave frequency. Utilizing 4 days of data they found the coefficient $X$ to be dependent on the radar's polarization, resulting in $X$=1.3 for vertical polarization and $X$=1.0 for horizontal polarization.

Within this paper, a new simple method is introduced to retrieve significant wave heights from X-band radar Doppler velocity measurements at near grazing incidence. The method is validated by wave measurements resulting from buoy as well as an acoustic wave and current profiler, which were both located within the range of the radar.

The paper is organized as follows. Section 2 introduces the utilized radar, the radar site as well as all additional data available for this study. In section 3 the methodology to retrieve the significant wave height from radar Doppler velocity records near grazing incidence is described. Within section 4 the method is validated by wave measurements from a buoy over a period of approximately 100 days. Furthermore, a discussion is given on the different sources of errors. Finally, conclusions and perspectives for future work are presented.





## 2 Experimental Setup and Data

All data used within this study were collected at the German Research Platform Fino-3, which is located 80 km west of the island of Sylt in the German Bight of the southern North Sea (Figure 1). The area within range of the radar (~3.2 km) has a water depth of approximately 22 m slightly increasing towards the northwest of the platform. For the predominant wind-generated young seas in the German Bight the area can be assumed to be of homogenous water depth and to the first order to be deep water with respect to the waves. The tidal range is about 1 m and ocean currents are mostly induced by semidiurnal tides with magnitudes below 0.6 m/s.

The utilized radar is a 12 kW marine X-band radar, which was modified to operate as a coherent-on-receive system, allowing to measure radar backscatter intensity and phase (Braun et al., 2008). The radar operates at 9.48 GHz with vertical polarization in transmit and receive (VV-pol). The pulse repetition frequency is 1 kHz with a pulse length of 50 ns, resulting in a range resolution of 7.5 m. The radar antenna has a vertical beam opening of 21° and a beam width of 7.5 feet (2.3 m) resulting in a horizontal (azimuth) resolution of ~1°. The signal runs through a linear amplifier and is digitized with 13 bits up to a maximum range distance of 3262.5 m. The radar can be operated with two different modes. In the rotational mode the antenna rotates at 30 rounds per minute capturing 360° of the surrounding of the platform (Figure 2). Within the static mode the antenna is oriented into a preselected direction where it then collects data over time. At Fino-3, the radar is mounted at a height of 43 m above the surface and was acquiring data for this study between 6. March and 14. July 2015. The radar was scheduled with an hourly cycle starting with 10 minutes of rotational data, which were utilized to retrieve the wave spectra and in particular the peak wave direction (Nieto-Borge et al., 1999). Within the following 32 minutes, 10 predefined directional scans were acquired in the static mode, which were not used within this study. After these 10 acquisitions the antenna was oriented into the radar-retrieved peak wave direction (looking up-wave) to acquire 15 minutes of data in the static mode. In addition to the radar data, a directional wave rider buoy Mark III of Datawell and an Acoustic Wave and Current Profiler (AWAC) from NORTEK were available within vicinity (< 300 m) of the Fino-3 platform, which were used for comparison and validation. In Figure 3 a timeseries of the environmental conditions throughout the analyzed time window is shown. Periods when the radar was not operating are highlighted in gray. In total, about 100 days of data are analyzed within this paper.

## 3 Methodology

To retrieve the Doppler speeds from the radar the so called pules-pair method is used (Zrnić, 1979), where the Doppler shift frequency $f_D$ is calculated using the derivative of the instantaneous phase $\phi_{el}$ of the coherent radar signal.

$$f_D = \frac{1}{2\pi}\frac{d\phi_{el}}{dt} = \frac{1}{2\pi}d\phi_{el}\,\text{PRF}, \tag{1}$$

where $t$ is time and PRF is the pulse repetition frequency of the radar. From those Doppler shift frequencies the corresponding radial Doppler speeds are calculated using





$$u_D = \frac{\lambda_{el}}{2\cos(\alpha)} f_D, \tag{2}$$

where $\lambda_{el}$ is the electromagnetic wave length of the radar (here $\lambda_{el} = 3.22$ cm). The PRF used for this study was 1000 Hz leading to a Doppler speed range of $\pm 8.05$ m/s. The cosine of the grazing angle $\alpha$ is approximately 1 as the measurements were acquired at low grazing incidence (here between 8.5° and 2.5°). To reduce the noise of the retrieved Doppler speeds,

the frequency shifts of 512 pulses are averaged, leading to an effective sampling frequency for the Doppler speeds of about 2 Hz.

When operating the radar in the rotational mode the retrieved Doppler speeds are fairly noisy and not well suited for investigation of surface waves. Therefore, the radar was operated in the static mode. To get the strongest contribution of surface waves to the radar Doppler speed measurements the radar beam was pointed into the peak wave direction and

operated in the static mode for 15 minutes to sample a sufficient number of wave groups. (Note that the number of observed wave groups defines the statistical variability of the estimated significant wave height.) A 250 s subsample of the extracted time-range map for the radar intensity and radial Doppler velocities is depicted in Figure 4. The modulation signal of the waves can be seen in the intensity as well as in the Doppler velocities. Furthermore, a well-known decrease of intensity and an increase of Doppler speed with range can be observed (for the later refer to Section 5). In Figure 5, time series of radar

intensity (A) and radial Doppler speeds (B) are plotted for the range distance of 525 m (Figure 4). For comparison, the heave measured by the directional wave rider buoy is plotted in Figure 4 C), which represents data that were recorded during the same time window but at a slightly different location (within a distance of <1 km). It can be seen that typical wave related features like wave groups are visible in both, the buoy heave time series and the Doppler velocity time series. Also the scales of such wave-related features correspond nicely between the radar and the buoy. However, at ranges above ~ 1000 m the

radar backscatter in the shadow of the wave crests is so low, that the values are close to or even at the noise floor of the radar and lead to uncertain Doppler speeds (Figure 4). Therefore, in the following, only data that were collected within a range of 300 to 1000 m of the radar are considered.

The aim of this study is to find a simple relation between the radar retrieved radial Doppler velocities and the significant wave height of the sea state. As the significant wave height is commonly calculated from the standard deviation of the

vertical displacement (heave), a linear regression analysis is carried out to find a relation between the standard deviation of radar Doppler velocities and buoy heave. In Figure 5, scatter plots are plotted for four range distances (375 m, 600 m, 712.5m and 825 m) showing the standard deviation of the heave measured by the buoy versus the standard deviation of the radar Doppler velocities. In all cases the offset $A$ and the slope $B$ are close to 0 and 1, respectively. With increasing range, there is a slight decrease of the offset (0.009 to 0.005) and a small increase of the slope (0.988 to 1.055). Therefore, the

relationship between the standard deviations of heave and Doppler velocity $\sigma_D$ can be simply assumed to be a one-to-one approximation. Taking this empirical relation, the significant wave height can be estimated from the Doppler record of 15 minutes at every range cell by simply using

$$H_{s_{radar}} = 4 \sqrt{\frac{1}{N} \sum_{i=1}^{N} \left( u_{D_i} - \bar{u}_D \right)^2} = 4 \, \sigma_D. \tag{3}$$



Note, that in this relation the units have to be adjusted to end up with the correct units for the significant wave height. To get a more stable estimate of Doppler radar retrieved significant wave height, the median is retrieved from all samples between 300 and 1000 m.

## 4 Results and Discussion

As shown in the previous section, a linear regression analysis shows that the standard deviation of the Doppler velocity time series is almost equal to the standard deviation of heave calculated from the wave rider buoy data. Therefore, the significant wave height can be estimated by simply calculating four times the standard deviation of the Doppler velocity. As mentioned above, only radar data acquired within a range of 300 to 1000 m were considered for the significant wave height estimate. For validation of the methodology the resulting radar retrieved significant wave heights are compared to results of a

directional wave rider as well as of a bottom mounted AWAC. Note that the significant wave height of the wave buoy uses a 30 min record and the AWAC a 10 minute time record, while the radar utilizes 15 min of data along a 700 m long transect. It should be noted that the considered sea states contain sufficient numbers of waves and wave groups, thus avoiding any significant bias. Figure 7 A) shows the scatter plot of significant wave heights resulting from the buoy versus those of the wave rider. For the statistical comparison 188 cases with very low backscatter (black ×) as well as 4 with very heavy rain

(pink +) were excluded from the total of 2654 data sets. The comparison resulted in a correlation coefficient of 0.96, a standard deviation of 0.23 m and a bias of 0.08m. Furthermore we compared the radar retrieved significant wave height to the AWAC results and the AWAC to the wave rider (Table 1), showing an overall excellent agreement of radar retrieved significant with those that can be obtained by well accepted measurements (~ 0.09 m). With this accuracy, the proposed method performs at least as well as the best results using traditional methods, however, without the need of any calibration

and sophisticated filtering techniques (Nieto-Borge et al. 1999; Vincent Bueno et al., 2012).
In the following, the results are compared to the method suggested by Hwang et al. (2010) and discussed with respect to physical explanation of this purely empirical relationship. For this, the geophysical interpretation of the Doppler signal from the ocean surface has to be discussed briefly. The radar-retrieved Doppler speed is a sum of multiple components i.e. wind drift, mean surface current, orbital motion of the waves, wave breaking as well as an incidence angle dependent component.

Wind drift and mean surface currents can be assumed to be constant within a time slot of the order of minutes. The orbital motion of the waves leads to a periodic modulation of the Doppler speed, which is mainly due to the horizontal orbital speeds of the surface waves. Assuming that wave breaking and the incidence angle dependence are small, linear wave theory can be applied to transform the orbital velocity spectrum to a wave amplitude spectrum. Hwang et al. (2010) showed that peak frequencies and wavelengths can be estimated reasonably well, while the integral spectral energy differs considerably

from the one retrieved by a buoy. The authors attributed this to various non-trivial uncertainties including directional distribution, shadowing effect, radar look direction with respect to wave propagation, swell modification and difference between spatial and temporal measurements. Therefore they suggest the empirical relationship for significant wave height



($H_s = 4\,X\,u_{DRMS}/\omega_p$) with an empirical correction factor X and the peak radial frequency $\omega_p$ of the sea state for unit consistency. They had 4 days of data and only considered 5 s of radar data for their comparison. For comparison their relationship was applied to the entire dataset used in this study resulting in the scatterplot shown in Figure 7 B) with a correlation coefficient of 0.96, a standard deviation of 0.31m and a bias of -0.08 m. The calibration coefficient X is found to

5 be 0.82 for the VV antenna used here. Note, that Hwang et al. (2010) had to detrend their datasets using a Butterworth filter technique because of the extremely short duration of their radar records. This is not necessary for the 15 minute long records used in this study. However, Hwang et al. (2010) used the RMS of the Doppler velocity which, after their detrending, is expected to be close to the standard deviation, which is used here ($u_{DRMS} \cong \sigma_D$) because no trend is present in time domain. Apparently the wave heights obtained using Hwang's method are significantly overestimated in high sea states, due to the

10 fact that high significant wave heights during storm situations are also associated with large peak wave periods > 10 s. In those cases a division by $\omega_p$ ($> 1$ for $T_p > 2\pi$) strongly increases the radar estimated significant wave height.

For deep water conditions, a division by the radial frequency transfers radial speeds to amplitudes according first order wave theory. The reasons why better results are obtained by not dividing by $\omega_p$ are not trivial and requires further investigation, which is beyond the scope of this paper. However, a few very likely sources leading to this behavior will be discussed.

Periodic features of the measured radar Doppler velocities are not only influenced by wave orbital speeds but also by wave induced variations in the wind field (e.g., Belcher and Hunt, 1998; Buckley and Veron, 2016) and therefore a periodically changing wind drift (Peirson and Garcia 2008). Also wave breaking causes a significant, instantaneous increase in Doppler speeds (Lee et al., 1995), which will raise the standard deviation of the Doppler speeds. In young sea states, which are strongly forced by the wind, the amount of wave breaking is enhanced and therefore an increase of the standard deviation of

the Doppler speeds is expected.

In order to further understand possible sources of error, the wave age is plotted versus the error in significant wave height when compared to the buoy (Figure 8). Note, that for simplicity here, the wave age is defined as the ratio between the phase velocity of the waves at the spectral peak and the 10 minute mean wind speed measured at 30 m height. The figure reveals a tendency to an overestimation for young sea states where the wind forces the waves and the rate of wave breaking is

25 expected to be considerably higher. As mentioned before, wave breaking increases the variance in Doppler velocities. The color scale corresponds to the directional spreading of the sea state measured by the wave rider. For young sea states, where neither an overestimation nor an underestimation can be found, the directionality is tendentiously higher than for the rest of the dataset. This might be explained by the fact that the radar was pointed statically into the main wave direction and therefore for waves travelling in all other direction the variance is decreased due to projection effects. This will most likely

cause an underestimation of significant wave height for sea states with a large spread. Additionally, multi-modal seas are expected to influence the accuracy of the method, because the energy of the second wave system is not caught by the radar if the secondary peak wave direction differs strongly from the first. For older sea states (or long waves) an underestimation is expected because linear wave theory has not been applied to transform the horizontal orbital speeds to surface elevation.



## 5 Conclusions and Outlook

Previous work has shown that retrieval of significant wave height from incoherent X-band marine radars requires a lot of calibration for each individual setup. Within this study a simple methodology is presented for estimating significant wave heights from the Doppler information retrieved from coherent marine radars. To do so, the radar is first acquiring an

intensity image sequence in the rotating mode to retrieve spectral wave parameters, e.g. wave length, period and direction of the sea state. To estimate the significant wave height the radar is pointed into the peak wave direction and acquires the Doppler information over 15 minutes. Validations using a wave rider buoy and an AWAC have shown that calculating the significant wave height using the empirically found relation $H_s = 4\,\sigma_{u_D}$ gives an accuracy of 0.23 m with a negligible bias of 0.08 m. The validation dataset which covers over 100 days of measurements includes a large number of different

environmental conditions, which is a major difference from previous studies and shows the overall excellent performance of the simple method.

Analysis of the error dependence on the wave age shows a tendency for an overestimation of significant wave height in young, wind driven sea states and an underestimation for swell. Additionally, an increase in the directional spreading of the wave field leads to smaller radar-retrieved significant wave heights.

Future research will focus on a better understanding of the causes of additional features, which are not related to the orbital speeds of the waves. A reliable detection of wave breaking might help avoid unrealistically high Doppler standard deviations. Moreover, a consideration of the wave-coherent wind drift effect could also improve the accuracy of the method. Projection effects, which are unavoidable because of the directionality of the wave fields, will be addressed in the future by including the directional energy distribution identified by the rotating radar image sequences. Furthermore, examining the applicability

of the proposed methodology for shallow water regions with inhomogeneous bathymetries would reveal a huge potential for field investigations of wave energy dissipation or wave current interactions in complex coastal environments. To increase the range of the radar, in particular for coastal applications, the method has to be extended to grazing incidence by accounting for the regions with a backscatter too low for reliable Doppler speed retrieval.

**Acknowledgements**

The authors would like to kindly thank Dr. Christian Senet of the Federal Maritime and Hydrographic Agency (BSH), for provision of the buoy and AWAC data at Fino-3. This work was partially funded by Germanys Federal Ministry for Economic Affairs and Energy (BMWi) under award 0327533C and supported through the Coastal Observing System for Northern and Arctic Seas (COSYNA).



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




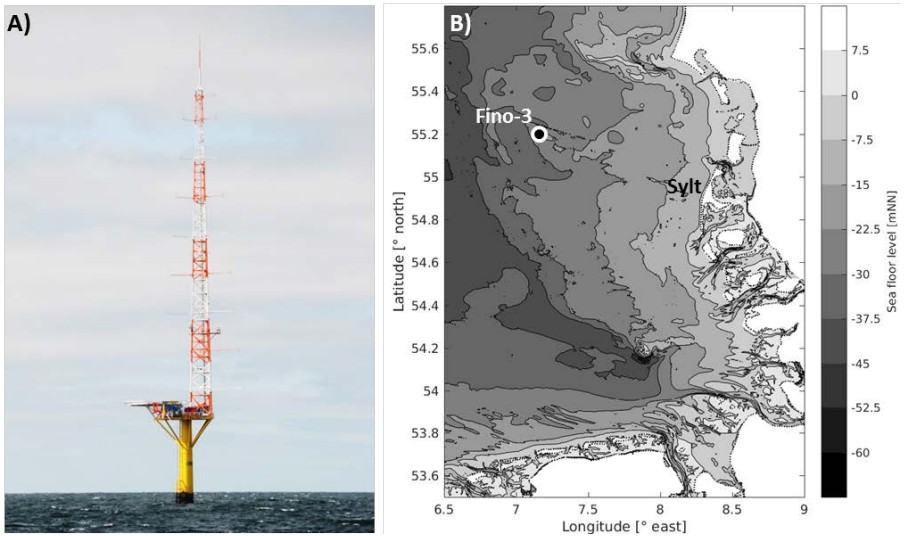

**Figure 1: Photograph of the research platform Fino-3, which is hosting the Dopplerized X-band marine radar at a height of 43 m. Fino-3 is located 80 km west off the island Sylt in the German Bight of the southern North Sea.**

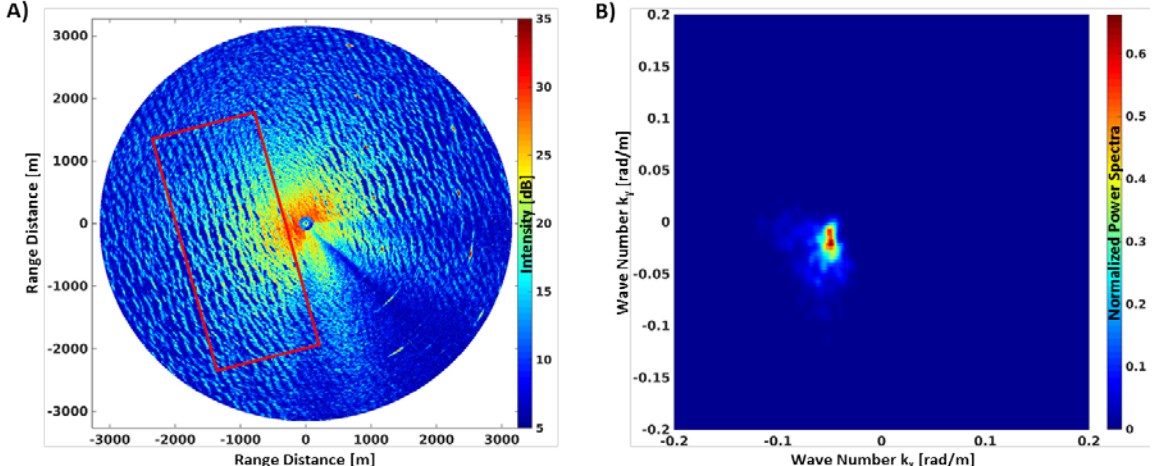

**Figure 2: Radar intensity image acquired in the polar mode at the research platform Fino-3 on 11. Aug 2014 at 10:00 UTC (A). Wave spectrum was retrieved from a 120 s long radar intensity image sequence (B).**



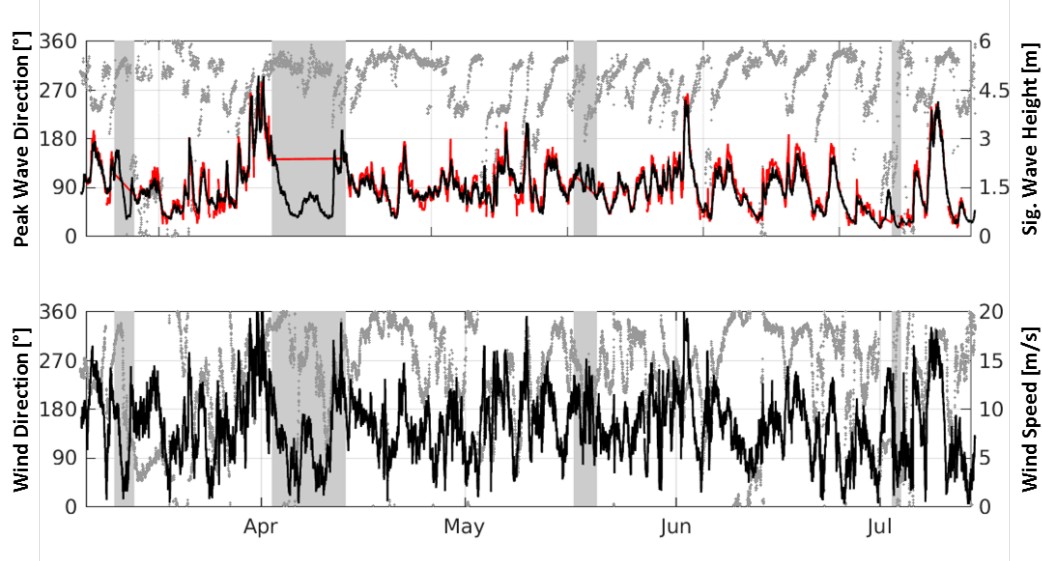

**Figure 3: Environmental conditions between 6. March and 14. July 2015 recorded at Fino-3. In the upper panel the grey line represents peak wave direction and the black and red line significant wave height from the buoy and radar respectively. In the lower panel the grey and black line give wind direction and wind speed.**

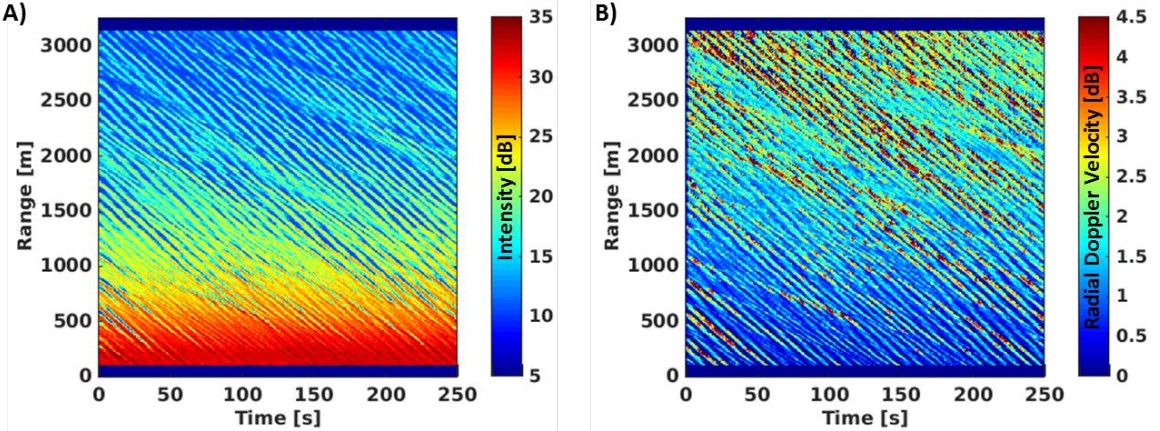

**Figure 4: Time-range plot showing radar data acquired in the static mode at Fino-3 on the 6 May 2015 at 23:44 UTC. Radar backscatter intensity (A) and radar retrieved radial Doppler velocity (B).**





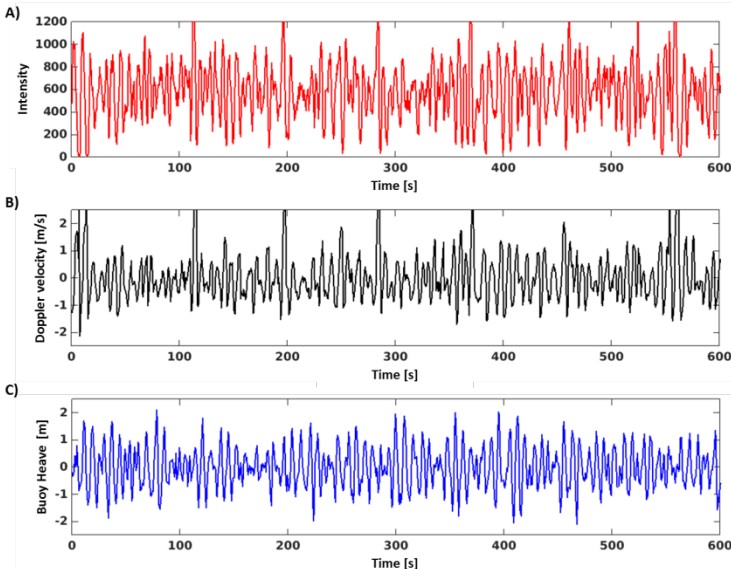

**Figure 5: Time series of radar intensity (A) and Doppler velocity (B) in a range distance of 525 m (Figure 4). Time series of surface heave recorded by the wave rider buoy (C) at the same time and located within vicinity of the radar measurements.**

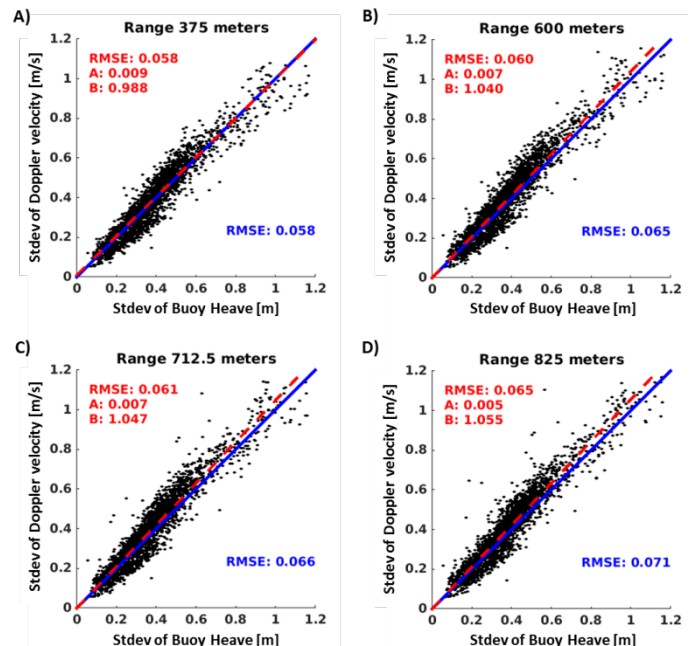

**Figure 6: Scatter plot of standard deviation (stdev) of buoy heave versus stdev radar Doppler velocity. The stdev of radar Doppler velocity was retrieved for range distances of A) 375 m, B) 600 m, C) 712.5m and D) 825 m. In the upper left of each plot the linear regression fit parameters A and B are given with $f(x) = A + Bx$.**

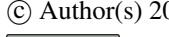


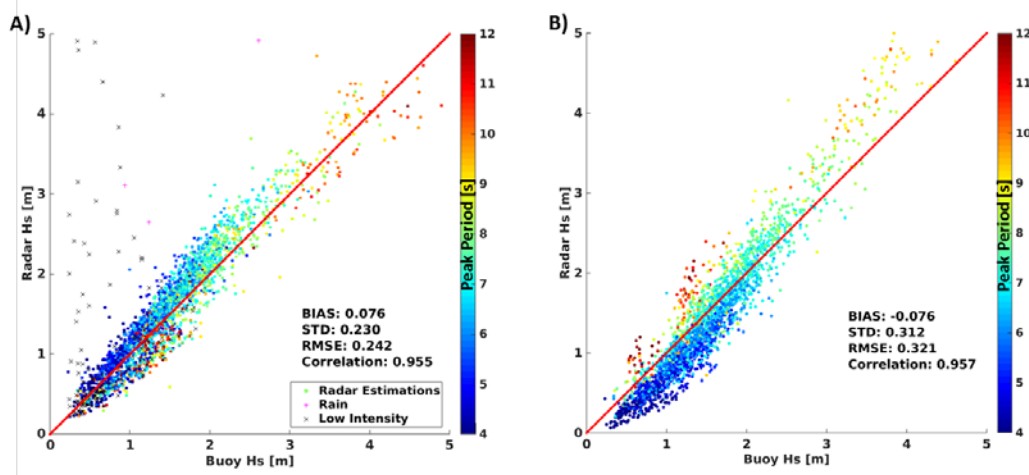

**Figure 7: Scatter plot of significant wave height from buoy data versus the significant wave height retrieved from the radar retrieved Doppler velocities using A) $H_s = 4\,\sigma_D$ and B) $H_s = 4 * 0.82\ u_{DRMS}/\omega_p$. Color coding gives the peak period resulting from the buoy data. A black x marks all radar data with very low radar backscatter, while the cases marked by a pink + are data which were recorded in very heavy rain. The statistics of the comparison, shown in the lower right, were retrieved excluding the cases with a low radar backscatter or which were acquired in heavy rain.**

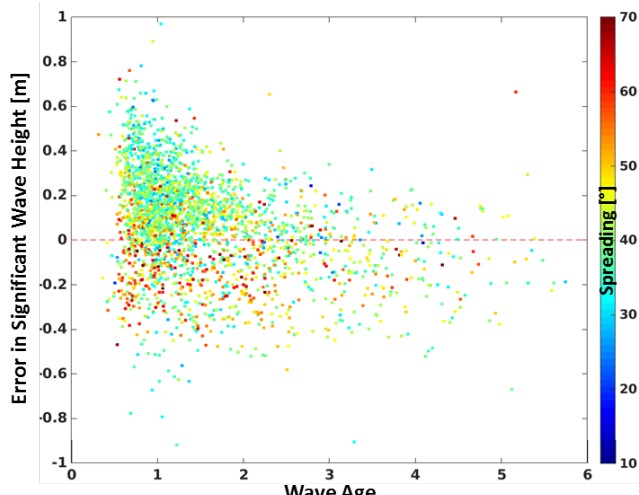

**Figure 8: Scatterplot of wave age versus error in significant wave height from comparison of the buoy to the radar. Color coding represents the spreading of the wave spectra.**



| | Buoy versus radar $H_s = 4\ \sigma_D$ | AWAC versus radar $H_s = 4\ \sigma_D$ | AWAC versus buoy | Buoy versus radar $H_s = 4 * 0.82\ u_{DRMS}/\omega_p$ |
|---|---|---|---|---|
| Correlation | 0.96 | 0.95 | 0,99 | 0.96 |
| RMSE [m] | 0.24 | 0.24 | 0,09 | 0.32 |
| Stdev [m] | 0.23 | 0.23 | 0,09 | 0.31 |
| Bias [m] | 0.08 | 0.08 | 0.01 | -0.08 |

**Table 1: Main statistical parameters resulting from comparisons.**