# Peer review of "A Simple Method for Retrieving Significant Wave Height from Dopplerized X-Band Radar"

_Ocean Science, 2016_

## Referee Comment (RC1) · Anonymous Referee #1 · 13 Sep 2016

The experimental results proof that the used coherent radar system is capable of estimate significant wave height properly. The results are convincing and, therefore, this work is suitable of being published.

There are some minor changes that the authors should be address:

Page 1. Line 10: the Doppler effect is induced by all the movement of the water surface, not only orbital velocities. Although from the obtained results it seems that the orbital velocities were dominant in the Doppler velocity measurements

Page 2. Line 4. The word "low" should be removed of this sentence, as, for grazing incidence conditions, the incidence angle has high values, close to 90 degrees.

[Figure]

Page 3. Equation 1. The last member of the equation should not have a differential of the phase, as it is multiplied by PRF. This member is an approximation of the member in the middle. For a better mathematical notation increment of the phase instead a differential should be written. In that case, the last member would contain an approximation of the derivative.

Page 6. The sentence in line 10 and 11 is not clear. Furthermore, the units of Tp are missing. Please, rewritten.

---

## Referee Comment (RC2) · B. Plant (Referee) · 20 Sep 2016

Review of "A simple method for retrieving significant wave height from Dopplerized X-band radar" by Carrasco et al., OS-2016-29.

This paper attempts to present a simple method for estimating significant wave height ($H_s$) from line-of-sight velocity ($V_{los}$) measurements using a pencil-beam, short-pulse, coherent-on-receive X-band radar oriented looking into the waves. To carry out this procedure correctly involves multiple difficulties, not the least of which is that swell and wind-waves do not usually propagate in the same direction. Nevertheless, the question being addressed here is whether one can obtain a reasonable estimate of $H_s$ from $V_{los}$. Unfortunately, the paper makes a very incomplete case that this can be done.

The true relationship between the wave-height-variance spectrum, $F_a$, and the variance spectrum of $V_{los}$, $F_V$, is

$$\int F_V(\omega, \varphi)d\omega d\varphi = \int \omega^2 F_a(\omega, \varphi - \varphi_a)d\omega d\varphi$$

where $\omega$ is angular frequency, $\varphi$ wave propagation direction relative to the wind and $\varphi_a$ is the antenna look direction relative to the wind. If one assumes that $F_a$ and $F_V$ are very sharply peaked at a given frequency and azimuth angle, then this may be written

$$F_V = \omega_p{}^2 F_a,$$

assuming that the antenna looks into the wave propagation direction. Therefore, in the authors' notation,

$$H_s = 4\sigma_D/\omega_p.$$

There is no doubt that $\omega_p$ belongs in the equation.

The problem, I think, is that in the real world it is not easy to determine $\omega_p$ because sometimes the frequency chosen as the peak of the spectrum is that of the wind waves and sometimes that of the swell. In reality, as we see above, $H_s$ is determined by the whole spectrum. Therefore, any simple method of determining it is bound to be approximate.

The authors' method may work better than one containing $\omega_p$ because of the difficulty of determining its value. However, their method is bound to be location-specific and the relationship will not always be $\omega_p = 1$ as the authors propose. Just think of carrying out their procedure in a wind wave tank at short fetch. (Yes, this can be done with a CW system and the antenna at a higher grazing angle.) The constant of proportionality will not be one.

The authors need to acknowledge this and show a histogram of $\omega_p$ for their entire time series. Care will have to be taken to be sure that $\omega_p$ corresponds to the type of waves carrying the most energy. The authors also need to inform the reader how the antenna was aligned to look into the waves and what was done when swell and wind waves were not aligned. Perhaps this was easy to do at the authors' site but it will not be so at all sites. Also, it would be nice to know which radar method was used to determine $H_s$ in Figure 8

---

## Referee Comment (RC3) · Anonymous Referee #3 · 12 Oct 2016

The paper describes a technique for estimating the significant wave height through the analysis of data acquired by a coherent radar. The technique is validated by a comparison with external measures.

The work is well written. In my opinion before it can be published authors should respond to some comments:

1) What happens in presence of a bimodal (or mor complex) waves? In other words could the authors explain how the results are affected by the angular spread of the wave spectrum?

2) line 15 pag. 3 do the authors refer to fig. 5C?

[Figure]

3) line 14 pag 5. 'radar' instead of 'ridar'.

4) line 15 p. 5 What is 0.09m?

---

## Author Comment (AC3) · 12 Oct 2016

The authors are thankful about the reviewers comments. We improved the manuscript based on the comments as follows (changes are highlighted in blue):

1. *"What happens in presence of a bimodal (or mor complex) waves?  In other words could the authors explain how the results are affected by the angular spread of the wave spectrum?"*

   As this important question is also pointed out by another reviewer, we modified the part of the manuscript that addresses this problem as follows:

[Figure]

Page 6, Lines 30-33: "Additionally, multi-modal seas are expected to influence the accuracy of the method, because the energy of the second wave system is not caught by the radar if the secondary peak wave direction differs strongly from the first. The energy of a second wave system travelling perpendicular to the antenna view direction would be underestimated due to a reduction of the radial velocity variations by projection effects. As the German Bight of the North Sea is not influenced by high energy swell events this might be a major issue in the open ocean where the presence of pronounced swell systems is more frequent. For older sea states (or long waves) an underestimation is expected because linear wave theory has not been applied to transform the horizontal orbital speeds to surface elevation."

2. *"line 15 pag. 3 do the authors refer to fig. 5C?"*
If it is correct that this comment refers to Page 4, Line 16, yes, we refer to Figure 5 C). This was corrected in the revised manuscript.

3. *"line 14 pag 5. 'radar' instead of 'ridar'"*
Error was changed to: "... scatter plot of significant wave heights resulting from the buoy versus those of the radar".

4. *"line 15 p. 5 What is 0.09m?"*
The number was a relict from a previous version where it referred to the AWAC vs. buoy RMSE. Thank you for pointing out this error. We modified this part:
"Furthermore we compared the radar retrieved significant wave height to the AWAC results and the AWAC to the wave rider (Table 1), showing an overall excellent agreement of radar retrieved significant wave heights with those that can be obtained by well accepted measurements (RMSE approx. 0.24 m)"

---

## Author Comment (AC4) · 17 Oct 2016

The following text and figure are an addition to the author comment AC2.

[Figure]

Figure 2: We have added to the histogram (shown in Figure 1) the root mean square error and bias of the significant wave height retrieved via the method of Huang et al. (green and blue) and our empirical approach (red and black).

**Fig. 1.**

---

## Author Response (AR1)

**Authors' Response to Referee Comments on "A Simple Method for Retrieving Significant Wave Height from Dopplerized X-Band Radar" by Ruben Carrasco et al.**

Michael Streßer
(on behalf of the authors)

October 13, 2016

**Answers to comments by Anonymous Referee #1**

The authors want to thank the anonymous referee #1 for his comments. We revised the manuscript as follows (colors refer to removed and added or changed text):

Referee comment: *Page 1. Line 10: the Doppler effect is induced by all the movement of the water surface, not only orbital velocities. Although from the obtained results it seems that the orbital velocities were dominant in the Doppler velocity measurements*

Authors response: The term 'radial velocity' here refers to the velocity of any scatterer in the view direction of the radar, sometimes also called 'line-of-sight' velocity. To avoid the chance to mix up the radial velocity with the wave orbital velocity the following was added in the revised manuscript:

Change: *In contrast to the backscatter intensity, the Doppler velocity measured by a coherent radar is induced by the radial velocity (or line-of-sight velocity) of the surface scattering and its periodic component is mainly the contribution of surface waves.*

Referee comment: *Page 2. Line 4. The word "low" should be removed of this sentence, as, for grazing incidence conditions, the incidence angle has high values, close to 90 degrees.*

Change: Page 2. Line 3. *Towards grazing incidence (high incidence angles $> 85\,°$) additional scattering mechanisms ...*

Referee comment: *Page 3. Equation 1. The last member of the equation should not have a differential of the phase, as it is multiplied by PRF. This member is an approximation of the member in the middle. For a better mathematical notation increment of the phase instead a differential should be written. In that case, the last member would contain an approximation of the derivative.*

Change: Page 3. Equation 1.

$$f_D = \frac{1}{2\pi}\frac{\partial \Phi_{el}}{\partial t} \approx \frac{1}{2\pi}(\Phi_{el,i+1} - \Phi_{el,i})\ PRF \qquad (1)$$

Referee comment: *Page 6. The sentence in line 10 and 11 is not clear. Furthermore, the units of Tp are missing. Please, rewritten.*

Change: Page 6, Lines 9-11. *Apparently the wave heights obtained using Hwang's method are significantly overestimated in high sea states, due to the fact that high significant wave heights during storm situations are also associated with large peak wave periods > 10 s. In those cases a division by $\omega_p$ (< 1 for $T_p$ > 2$\pi$) strongly increases the radar estimated significant wave height because $\omega_p$ is smaller than one for long wave periods ($\omega_p$ < 1 rad/s for $T_p$ > 2$\pi$ s).*

**Answers to comments by Bill Plant**

The authors appreciate the thorough and constructive review by Dr. Plant. When we started analyzing the dataset we were also very surprised about the outcome of the statistical results. We are aware of the dependencies which we neglected and therefore we welcome a scientific discussion on the issues that are pointed out by Dr. Plant.

Referee comment: Dr. Plant depicts in his comments the physical relationship between the line-of-sight velocities and the height of ocean surface waves:

*"The true relationship between the wave-height-variance spectrum, $F_a$, and the variance spectrum of $V_{los}$, $F_V$, is*

$$\int F_V(\omega, \varphi)d\omega d\varphi = \int \omega^2 F_a(\omega, \varphi - \varphi_a)d\omega d\varphi \qquad (2)$$

*where $\omega$ is angular frequency, $\varphi$ wave propagation direction relative to the wind and $\varphi_a$ is the antenna look direction relative to the wind. If one assumes that $F_a$ and $F_V$ are very sharply peaked at a given frequency and azimuth angle, then this may be written*

$$F_V = \omega_p^2 F_a, \qquad (3)$$

*assuming that the antenna looks into the wave propagation direction. There-*
*fore, in the authors' notation,*

$$H_s = 4\sigma_D/\omega_p. \tag{4}$$

*There is no doubt that $\omega_p$ belongs in the equation."*

Authors response: The authors are aware of this relationship and agree
that from the physical point of view, according to linear wave theory, the
radial frequency has to be taken into account to transform from speeds to
heave. However, we realized that a simple, empirically derived relationship
between the periodic features in the Doppler signal and the Significant Wave
Height (Hs) of the sea state is statistically performing better than previous
attempts of estimating Hs using physically more substantiated approaches.
This new finding was the main motivation for the submission of this article.

Referee comment: *"In the real world it is not easy to determine $\omega_p$ be-
cause sometimes the frequency chosen as the peak of the spectrum is that of
the wind waves and sometimes that of the swell. In reality, as we see above,
$H_s$ is determined by the whole spectrum. Therefore, any simple method of
determining it is bound to be approximate.*

*The authors' method may work better than one containing $\omega_p$ because of
the difficulty of determining its value. However, their method is bound to be
location-specific and the relationship will not always be $\omega_p = 1$ as the authors
propose. Just think of carrying out their procedure in a wind wave tank at
short fetch. (Yes, this can be done with a CW system and the antenna at a
higher grazing angle.) The constant of proportionality will not be one.*

*The authors need to acknowledge this and show a histogram of $\omega_p$ for
their entire time series. Care will have to be taken to be sure that $\omega_p$ corre-
sponds to the type of waves carrying the most energy."*

Authors response: Indeed it is a hard task to assess a representative esti-
mate of $\omega_p$ for a specific wave spectrum. For most of the cases of this study,
the highest energy peak is clearly defined. This might be due to the fact
that the German Bight of the North Sea is not influenced by high energy
swell events and we think that for the future it would be great to check the
applicability of the method to swell-dominated areas. Nevertheless, the ana-
lyzed dataset contains a high variation of the peak frequency. Fig. 1 (of this

document) shows a histogram of $\omega_p$ for the whole dataset used in this study. It can be seen that, the peak radial frequencies range from approximately 0.25 up to 2.5 with most events slightly smaller than 1. The reason why we decided not to include this figure in the article is that we think that the color-coding in the scatter plot depicted in Figure 7 delivers enough information on the variability of the Peak Period for the analysed dataset. In addition, Figure 3 provides information on the large variety of different wave conditions, from developing seas to storm events as well as decaying seas.

[Figure]

Figure 1: Histogram of peak radial frequencies for the analysed dataset

Referee comment: *"The authors also need to inform the reader how the antenna was aligned to look into the waves and what was done when swell and wind waves were not aligned. Perhaps this was easy to do at the authors' site but it will not be so at all sites."*

Authors response: The radar was scheduled as described here (Page 3, Lines 13-21). We added a line to the revised manuscript (highlighted in blue) to inform the reader about what was done in swell cases:

Change: "The radar can be operated with two different modes. In the rotational mode the antenna rotates at 30 rounds per minute capturing 360° of the surrounding of the platform (Figure 2). Within the static mode the

antenna is oriented into a preselected direction where it then collects data over time. ... The radar was scheduled with an hourly cycle starting with 10 minutes of rotational data, which were utilized to retrieve the wave spectra and in particular the peak wave direction (Nieto-Borge et al., 1999). Within the following 32 minutes, 10 predefined directional scans were acquired in the static mode, which were not used within this study. After these 10 acquisitions the antenna was oriented into the radar-retrieved peak wave direction (looking up-wave) to acquire 15 minutes of data in the static mode. For multi-modal sea states the radar antenna was solely pointing in the direction of the highest energy peak of the radar image spectra derived automatically from the polar radar image sequences."

Referee comment:    *"... Perhaps this was easy to do at the authors' site but it will not be so at all sites."*

Authors response: Of course this might be a limitation of the proposed method for cases with a significant portion of energy located in a second wave system traveling perpendicular to the main one. As mentioned above the German North Sea is weakly influenced by strong swell events. Therefore, if a second system is apparent in the analysed dataset, its energy is mostly small (amplitudes $< 0.5$ m). We thank Dr. Plant for pointing out the need for a more profound discussion on that topic and modified our consideration of this limitation in the Results and Discussion section of the paper:

Change: Page 6, Lines 30-33: "Additionally, multi-modal seas are expected to influence the accuracy of the method, because the energy of the second wave system is not caught by the radar if the secondary peak wave direction differs strongly from the first. The energy of a second wave system travelling perpendicular to the antenna view direction would be underestimated due to a reduction of the radial velocity variations by projection effects. As the German Bight of the North Sea is not influenced by high energy swell events this might be a major issue in the open ocean where the presence of pronounced swell systems is more frequent. For older sea states (or long waves) an underestimation is expected because linear wave theory has not been applied to transform the horizontal orbital speeds to surface elevation."

*"Also, it would be nice to know which radar method was used to determine Hs in Figure 8."*

It's the simple method introduced in this paper. We have added the information in the figure caption.

**Answers to comments by Anonymous Referee #3**

The authors are thankful about the reviewers comments. We improved the manuscript based on the comments as follows (changes are highlighted in blue):

Referee comment: *"What happens in presence of a bimodal (or mor complex) waves? In other words could the authors explain how the results are affected by the angular spread of the wave spectrum?"*

Author response: As this important question is also pointed out by another reviewer, we modified the part of the manuscript that addresses this problem as follows:

Change: Page 6, Lines 30-33: "Additionally, multi-modal seas are expected to influence the accuracy of the method, because the energy of the second wave system is not caught by the radar if the secondary peak wave direction differs strongly from the first. The energy of a second wave system travelling perpendicular to the antenna view direction would be underestimated due to a reduction of the radial velocity variations by projection effects. As the German Bight of the North Sea is not influenced by high energy swell events this might be a major issue in the open ocean where the presence of pronounced swell systems is more frequent. For older sea states (or long waves) an underestimation is expected because linear wave theory has not been applied to transform the horizontal orbital speeds to surface elevation."

Referee comment: *"line 15 pag. 3 do the authors refer to fig. 5C?"*

Author response: If it is correct that this comment refers to Page 4, Line 16, yes, we refer to Figure 5 C).

Change: This was corrected in the revised manuscript.

Referee comment: *"line 14 pag 5. 'radar' instead of 'ridar'"*

Author Response: Error was changed to:

Change: "... scatter plot of significant wave heights resulting from the buoy versus those of the radar".

Referee comment: *"line 15 p. 5 What is 0.09m?"*

Author Response: The number was a relict from a previous version where it referred to the AWAC vs. buoy RMSE. Thank you for pointing out this error. We modified this part:

[revised manuscript text omitted]
 because $\omega_p$ is smaller than one for long wave periods ($\omega_p < 1 \frac{rad}{s}\ for\ T_p > 2\pi\ s$).

For deep water conditions, a division by the radial frequency transfers radial speeds to amplitudes according first order wave theory. The reasons why better results are obtained by not dividing by $\omega_p$ are not trivial and requires further investigation, which is beyond the scope of this paper. However, a few very likely sources leading to this behavior will be discussed. Periodic features of the measured radar Doppler velocities are not only influenced by wave orbital speeds but also by wave induced variations in the wind field (e.g., Belcher and Hunt, 1998; Buckley and Veron, 2016) and therefore a periodically changing wind drift (Peirson and Garcia 2008). Also wave breaking causes a significant, instantaneous increase in Doppler speeds (Lee et al., 1995), which will raise the standard deviation of the Doppler speeds. In young sea states, which are

strongly forced by the wind, the amount of wave breaking is enhanced and therefore an increase of the standard deviation of the Doppler speeds is expected.

In order to further understand possible sources of error, the wave age is plotted versus the error in significant wave height when compared to the buoy (Figure 8). Note, that for simplicity here, the wave age is defined as the ratio between the phase velocity of the waves at the spectral peak and the 10 minute mean wind speed measured at 30 m height. The figure reveals a tendency to an overestimation for young sea states where the wind forces the waves and the rate of wave breaking is expected to be considerably higher. As mentioned before, wave breaking increases the variance in Doppler velocities. The color scale corresponds to the directional spreading of the sea state measured by the wave rider. For young sea states, where neither an overestimation nor an underestimation can be found, the directionality is tendentiously higher than for the rest of the dataset. This might be explained by the fact that the radar was pointed statically into the main wave direction and therefore for waves travelling in all other direction the variance is decreased due to projection effects. This will most likely cause an underestimation of significant wave height for sea states with a large spread. Additionally, multi-modal seas are expected to influence the accuracy of the method, because the energy of the second wave system is not caught by the radar if the secondary peak wave direction differs strongly from the first. The energy of a second wave system travelling perpendicular to the antenna view direction would be underestimated due to a reduction of the radial velocity variations by projection effects. As the German Bight of the North Sea is not influenced by high energy swell events this might be a major issue in the open ocean where the presence of pronounced swell systems is more frequent. 
[revised manuscript text omitted]